# Palbociclib Promotes Dephosphorylation of NPM/B23 at Threonine 199 and Inhibits Endometrial Cancer Cell Growth

**DOI:** 10.3390/cancers11071025

**Published:** 2019-07-20

**Authors:** Chiao-Yun Lin, Li-Yu Lee, Tzu-Hao Wang, Cheng-Lung Hsu, Chia-Lung Tsai, Angel Chao, Chyong-Huey Lai

**Affiliations:** 1Department of Obstetrics and Gynecology, Chang Gung Memorial Hospital Linkou Medical Center and Chang Gung University College of Medicine, Taoyuan 333, Taiwan; 2Gynecologic Cancer Research Center, Chang Gung Memorial Hospital, Taoyuan 333, Taiwan; 3Department of Pathology, Chang Gung Memorial Hospital Linkou Medical Center, Taoyuan 333, Taiwan; 4Genomic Medicine Research Core Laboratory, Chang Gung Memorial Hospital, Taoyuan 333, Taiwan; 5Department of Medical Oncology, Chang Gung Memorial Hospital Linkou Medical Center, Taoyuan 333, Taiwan

**Keywords:** endometrial cancer, palbociclib, megesterol acetate, combination treatment, NPM/B23, Thr199

## Abstract

Endometrial cancer incidence rates are growing, especially in countries with rapid socioeconomic transitions. Despite recent advances in chemotherapy, hormone therapy, and targeted therapy, advanced/recurrent disease remains a clinical challenge. Palbociclib—a selective inhibitor of cyclin-dependent kinases (CDK) 4/6—has therapeutic potential against estrogen receptor (ER)-positive and HER2-negative breast cancer. However, the question as to whether it can be clinically useful in endometrial cancer remains open. Here, we show that combined treatment with palbociclib and megesterol acetate exerts synergistic antiproliferative effects against endometrial cancer cells. Treatment of cancer cells with palbociclib suppressed NPM/B23 phosphorylation at threonine 199 (Thr199). We further demonstrated that CDK6 acts as a NPM/B23 kinase. Palbociclib-induced NPM/B23 dephosphorylation sensitized endometrial cancer cells to megesterol acetate through the upregulation of ERα expression. Immunohistochemistry revealed an overexpression of phospho-NPM/B23 (Thr199) in human endometrial cancer, and phospho-NPM/B23 (Thr199) expression levels were inversely associated with Erα in clinical specimen. In a xenograft tumor model, the combination of palbociclib and megesterol acetate successfully inhibited tumor growth. Taken together, our data indicate that palbociclib promoted NPM/B23 dephosphorylation at Thr199—an effect mediated by disruption of CDK6 kinase activity. We conclude that palbociclib holds promise for the treatment of endometrial cancer when used in combination with megesterol acetate.

## 1. Introduction

Endometrial cancer is the sixth most common gynecologic cancer worldwide and its incidence rate is growing—especially in countries with rapid socioeconomic transitions [1]. Between 2014 and 2015, Taiwan has witnessed a 7.73% increase in the incidence of this malignancy, with 20.71 new cases per 100,000 persons/year [2]. Although hysterectomy remains the mainstay of definitive therapy [3], adjuvant chemo- and/or radio-therapy may confer a survival benefit in patients who are at high risk for recurrences or presenting with advanced disease. Unfortunately, the clinical management of advanced/recurrent disease is still challenging and the role of chemotherapy, hormone therapy, and targeted therapy remains under scrutiny.

Early endometrial cancers generally express estrogen and progesterone receptors [4]. Patients with progesterone receptor (PR)-positive as well as estrogen receptor (ER)-positive responded to medroxyprogesterone acetate, a progestin [5]. Furthermore, the single best predictor of response was ER, which was significantly associated with clinical response and survival [6]. A reduced expression of ER has been associated with poor differentiation, advanced-stage tumors, disease recurrence, and adverse outcomes [7]. Meanwhile, the disappearance of hormonal receptor expression is common in patients with recurrent estrogen-associated cancers [8]. It is critical to find new therapeutic targets that can restore functional hormones through either PR or ER, which sensitize to hormone therapy.

During early mitosis, the nucleolus starts to disassemble—an event accompanied by phosphorylation of cyclin-dependent kinase (CDK) [9]. NPM/B23 (nucleophosmin, NPM, B23)—one of the most abundant nucleolar proteins—undergoes CDK-mediated phosphorylation [9]. During cell cycle, NPM/B23 can be phosphorylated at different sites [10]. Phosphorylation at the casein kinase 2 (CK2) site (Ser125) regulates NPM/B23 function in the nucleolus [11], whereas phosphorylation at CDK1 sites, including threonine (Thr) 199, 219, 234, and 237, promotes dissociation of NPM/B23 from the nucleolus [12]. NPM/B23 dissociation from the centrosome at the beginning of cell division is mediated by CDK2/cyclin E phosphorylation at Thr199 [13]. However, NPM/B23 hyperphosphorylation at Thr199 deregulates both CDK2 and CDK4, ultimately resulting in an abnormal centrosome duplication and the disruption of genome integrity [14]. Interestingly, the v-cyclin and cellular CDK6 kinase of Kaposi’s sarcoma herpesvirus is capable of phosphorylating NPM/B23 on Thr199, with phosphorylated NPM/B23 being detectable in primary Kaposi’s sarcoma [15]. Moreover, CDK1/cyclin B can induce NPM/B23 phosphorylation at Thr234 and Thr237, which promotes its re-association to the centrosome during mitosis [13].

An altered cell proliferation and the capability to escape immune-mediated clearance are common features of malignant cells [16]. At the molecular level, dysregulation of the cyclin D-CDK 4/6-inhibitor of the CDK4 (INK4)-retinoblastoma (Rb) pathway is frequently observed in cancer [17]. The cyclin D-CDK4/6-INK4-Rb pathway is the master regulator of the G1-S transition during the cell cycle. Inhibition of CDK4/6 exerts multiple biological effects, including (1) decreased levels of phosphorylated Rb, (2) increased formation of Rb-E2F complexes, and (3) inhibition of E2F transcription factor activity. CDK4/6 inhibitors (i.e., palbociclib, abemaciclib, and ribociclib) have been granted approval from the the U.S. Food and Drug Administration for the treatment of patients with previously untreated estrogen receptor (ER)-positive advanced breast malignancies [18,19,20,21,22]. Notably, palbociclib has shown promising therapeutic activity against endometrial cancer cells [23] and PTEN-deficient endometrial malignant cells [24].

We have previously shown that estrogens may stabilize NPM/B23 [25]. We have also demonstrated that NPM/B23 silencing via AP2γ (activator protein-2γ, TFAP2C) may activate ERα expression in endometrial cancer cells [26]. Knockdown of AP2γ in an animal model sensitized endometrial cancer cells to megesterol acetate through an upregulation of ERα expression. Unfortunately, NPM/B23 or AP2γ inhibitors are not currently available for clinical use. In this context, we reasoned that palbociclib, a CDK4/6 inhibitor that is capable of regulating the cell cycle machinery [27], could modulate ERα expression in endometrial cancer cells by affecting the NPM/B23 phosphorylation status.

Here, we show that combined treatment with palbociclib and megesterol acetate exerts synergistic antiproliferative effects against endometrial cancer cells. Exposure of cancer cells to palbociclib suppressed NPM/B23 phosphorylation at Thr199. We further demonstrated that CDK6 acts as a NPM/B23 kinase. Palbociclib-induced NPM/B23 dephosphorylation sensitized endometrial cancer cells to megesterol acetate through upregulation of ERα expression.

## 2. Results

### 2.1. Exposure to Palbociclib Induced ERα Expression in Endometrial Cancer Cells

Because palbociclib is a selective inhibitor of cyclin-dependent kinases CDK4 and CDK6 and NPM/B23 can be phosphorylated either by CDK4 [14] or CDK6 [15], we tested whether this drug was able to inhibit NPM/B23 phosphorylation at several sites. Upon treatment of ERα-negative (ARK2, high-passage Ishikawa cells [26]) and ERα-positive HEC1B cells with palbociclib, cell lysates were probed with phospho specific NPM/B23 antibodies. Notably, palbociclib enhanced ERα expression and promoted NPM/B23 dephosphorylation at threonine 199, 234, and 237, whereas NPM/B23 dephosphorylation at serine 125 and PR expression was not significantly altered (Figure 1 and Appendix A). Exposure to palbociclib also induced the expression of several ERα-regulated genes, including *cathepsin D*, *EBAG9*, and *TFF1/pS2* (Figure 1c). Taken together, these findings indicate that palbociclib is able to specifically inhibit NPM/B23 phosphorylation at Thr199, 234, and 237.

### 2.2. Palbociclib and Megestrol Acetate Synergistically Inhibit Survival, Increase Apoptosis, and Increase the Expression of ERα in Endometrial Cancer Cells

Because restoration of ERα expression renders endometrial cancer cells susceptible to megestrol acetate treatment [26], we investigated whether combined treatment with palbociclib and megestrol acetate could exert synergistic antiproliferative effects. Chou-Talalay plots based on the MTT assay revealed that palbociclib and megestrol acetate synergistically inhibited cell viability (Figure 2a,b; Appendix A). The combination of palbociclib and megestrol acetate also showed a synergistic effect on colony formation (Figure 2c and Appendix A) and induction of apoptosis—as reflected by higher levels of cleaved PARP (Figure 2d). The results of tumor xenograft experiments confirmed that palbociclib and megestrol acetate used in combination inhibited tumor growth to a greater extent than either drug alone (Figure 2e and Appendix A). At the mechanistic level, palbociclib was found to specifically inhibit phospho-NPM/B23 (Thr199) and was able to upregulate ERα expression (Figure 2f). The combination of palbociclib and letrozole also showed synergistic antiproliferative effects against tumor cell growth (Appendix A).

### 2.3. Phospho-NPM/B23 (Thr199) Is Involved in Endometrial Tumorigenesis

To further investigate the role of phospho-NPM/B23 (Thr199) and phospho-NPM/B23 (Thr234/237) in endometrial cancer, we examined their immunohistochemical expression in pathological specimens obtained from premenopausal women. Phospho-NPM/B23 (Thr199) histoscores were significantly higher in endometrial cancer than in normal endometrial tissue (*p* = 0.020), whereas differences were less obvious for phospho-NPM/B23 (Thr234/237) (Figure 3a and Appendix A). Representative images of phospho-NPM/B23 (Thr199) expression in malignant and normal endometrial tissue are provided in Figure 3b.

To shed more light on the impact of NPM/B23 dephosphorylation and phosphorylation on ERα expression, HEC1B cells were transiently transfected with wild-type NPM/B23 (WT), a dephospho-mimetic NPM/B23 mutant (Thr199A), or a phospho-mimetic NPM/B23 mutant (Thr199D). Overexpression of the dephospho-mimetic NPM/B23 mutant (Thr199A) promoted ERα expression at both mRNA and protein levels (Figure 3c,d). Moreover, phospho-NPM/B23 (Thr199) histoscores were significantly higher in ERα negative endometrial cancer specimens than in ERα positive patients (*p* = 0.026; Appendix A). Conversely, overexpression of the phospho-mimetic NPM/B23 mutant (Thr199D) promoted both cell proliferation and colony formation (Figure 3e,f). All of the subsequent experiments were therefore focused on phospho-NPM/B23 (Thr199). These results confirmed that phospho-NPM/B23 (Thr199) expression levels were inversely associated with ERα in clinical specimen.

### 2.4. CDK6-Mediated Phosphorylation of NPM/B23 Promotes ERα Expression

We next sought to identify the kinase that specifically mediated palbociclib-induced NPM/B23 dephosphorylation. Because palbociclib is capable of inhibiting both CDK4 and CDK6, ablation of either enzyme was achieved through specific siRNAs. ARK2 and HEC1B cells in which expression of either CDK4 or CDK6 was silenced were characterized by reduced phospho-NPM/B23 (Thr199) and increased ERα expression (Figure 4). To further investigate whether CDK4 or CDK6 can directly interact with NPM/B23, immunoprecipitation experiments were performed. The results revealed that CDK6 was able to interact with NPM/B23 (Figure 5a), whereas CDK4 was not (Figure 5b). We therefore implemented an in vitro kinase assay to assess whether CDK6 is actually capable of phosphorylating NPM/B23. By monitoring phosphorylation levels of endogenous NPM/B23 in immunoprecipitates, we found that purified recombinant CDK6/cyclin D1 was able to add phosphate groups at the Thr199 residue of NPM/B23. Notably, this specific kinase activity was abrogated upon palbociclib treatment (Figure 5c). Taken together, these results indicate that (1) palbociclib is capable of inducing dephosphorylation of NPM/B23 at the Thr199 site and (2) CDK6 is involved in the phosphorylation of NPM/B23 in endometrial cancer.

### 2.5. Phosphorylated NPM/B23 (Thr199) Forms a Preferential Complex with CDK6

NPM/B23 was found to form a protein complex with CDK6 (Figure 5a). Notably, this interaction was significantly weakened upon treatment with palbociclib (Figure 6a). The association between NPM/B23 and CDK6 was further confirmed when Thr at position 199 was replaced with Ala in the phospho-mimetic mutant NPM/B23 (Thr199D) (Figure 6b). Altogether, these data indicate that the NPM/B23/CDK6 interaction critically depends on phosphorylation at the Thr199 residue.

## 3. Discussion

In this study, we demonstrated that an increased expression of phosphorylated NPM/B23 (at Thr199) in cancer cells promotes its interaction with CDK6 and plays a key role in endometrial tumorigenesis. By inhibiting the CDK4/6 kinase, palbociclib disrupts this interaction and promotes ERα expression, ultimately sensitizing hormone-refractory endometrial cancer cells to endocrine therapy (Figure 7).

The potential clinical usefulness of palbociclib in patients with endometrial malignancies remains unclear. A phase II clinical trial (NCT02730429) is currently examining the association of letrozole, an aromatase inhibitor, with palbociclib/placebo in patients with metastatic endometrial cancer, but results are still unavailable. Our current preclinical data indicate that the combination of palbociclib and letrozole can exert synergistic effects against tumor growth (Appendix A).

The PALOMA 2 phase III trial demonstrated that in previously untreated patients with ER-positive, HER2-negative advanced breast cancer, the combination of palbociclib and letrozole resulted in a significantly longer progression-free survival compared with letrozole alone (24.8 versus 14.5 months, respectively, *p* < 0.001) [18]. Moreover, the PALOMA 3 phase III trial showed that the progression-free survival of patients treated with palbociclib plus fulvestrant was significantly longer than that of patients who received fulvestrant alone (9.5 versus 4.5 months, respectively, *p* < 0.001) [19].

In our study, palbociclib and megestrol acetate were found to exert synergistic antiproliferative effects against ERα-negative ARK2. However, the results obtained in our xenograft model revealed that palbociclib alone was capable of reducing cell proliferation in ERα-positive HEC1B cells, but not in ERα-negative ARK2 cells (Figure 2 and Appendix A). The key relevance of ERα in mediating palbociclib activity is consistent with literature data obtained on breast cancer [28]. Our data point to a therapeutic utility of palbociclib plus megestrol acetate either in ERα-positive endometrial cancer or for reversing hormone therapy resistance in Erα-negative endometrial tumors. We postulated that the effect of combined treatment would be the activation of Erα by palbociclib, which leads to growth inhibition by megestrol acetate. Because palbociclib is not a specific NPM/B23 inhibitor, we cannot rule out other mechanisms that activate ERα to result in decreased tumor growth by megestrol.

NPM/B23 is a nuclear protein found to be overexpressed in several malignancies [13]. Interestingly, dephosphorylation of NPM/B23 has been shown to result in restoration of E2F1-dependent DNA repair [29]. Phosphorylation of the Thr234/237 sites in hepatocellular carcinoma cells was related to invasiveness and migration [30]. Phosphorylation of NPM/B23 at Thr199 is mediated by CDK2/cyclin E, with phospho-NPM/B23 (Thr199) being involved in both centrosome duplication and pre-mRNA processing [31]. We showed that phospho-NPM/B23 (Thr199) is a substrate of CDK6-cyclin D1 and phosphorylation of NPM/B23 is a prerequisite for CDK6 interaction.

Numerous studies have identified factors that may influence palbociclib activity, including Rb [22], hormone receptor status [18], p16 expression [32], and PTEN [24]. Our current data indicate that palbociclib treatment reduced phospho-NPM/B23 (Thr199) and induced ERα expression both in human endometrial cancer cells and in a xenograft tumor model. Moreover, IHC revealed that phospho-NPM/B23 (Thr199) is overexpressed in endometrial cancer, suggesting that it could serve as a biomarker to guide palbociclib treatment in future clinical trials.

NPM/B23 has the potential to act as an oncoprotein and may play a role in targeted cancer therapy. Several molecules that are capable of interacting with NPM/B23 have shown promising antiproliferative potential [33], albeit they are unsuitable for drug development. Although palbociclib is not a direct inhibitor of NPM/B23, it can to some extent disrupt its interaction with CDK6.

## 4. Materials and Methods

### 4.1. Cell Cultures

HEC1B endometrial cancer cells were purchased from the American Type Culture Collection (Manassas, VA, USA). ARK2 endometrial cancer cells were obtained from Dr. Santin (Yale University, School of Medicine, New Haven, CT, USA) [34]. Ishikawa endometrial cancer cells were kindly provided by Dr. Nishida (National Hospital Organization, Kasumigaura Medical Center, Japan). HEC1B cells and Ishikawa cells were grown in α-MEM medium containing 15% (*v*/*v*) fetal bovine serum (FBS). ARK2 cells were grown in RPMI-1640 medium containing 10% (*v*/*v*) FBS.

### 4.2. Antibodies, Reagents, and Plasmids

Mouse monoclonal antibodies against NPM/B23, β-actin, CDK4, and CDK6 were obtained from Santa Cruz Biotechnology (Santa Cruz, CA, USA). Rabbit polyclonal antibodies against PARP, ERα, and phospho-NPM/B23 (Thr234/237) and rabbit monoclonal antibodies raised against phospho-NPM/B23 (Ser125) and HA were purchased from Abcam (Cambridge, MA, USA). The rabbit polyclonal antibody against phospho-NPM (Thr199) was from Cell Signaling Technology (Danvers, MA, USA). Further, anti-PR rabbit monoclonal antibody was obtained from GeneTex (San Antonio, TX, USA); Palbociclib and megestrol acetate were obtained from Pfizer Manufacturing Deutschland GmbH (Freiburg, Germany) and TTY Biopharm Company (Taipei City, Taiwan), respectively. All chemicals were from Sigma (St. Louis, MO, USA) unless otherwise indicated. The expression vector plasmids for HA-CDK4 and HA-CDK6 were from Addgene (Cambridge, MA, USA). The expression vector plasmids for NPM/B23 (WT), NPM/B23 (Thr199A), and NPM/B23 (Thr199D) were previously described in detail [29].

### 4.3. DNA Transfection Experiments

In overexpression experiments, NPM/B23 (WT), NPM/B23 (Thr199A), NPM/B23 (Thr199D), HA-CDK4, HA-CDK6, and pcDNA (V) were transfected into ARK2 cells using the Lipofectamine 2000 reagent (Invitrogen, Carlsbad, CA, USA) according to the manufacturer’s protocol. Transient transfection was performed 24 h before cell harvesting.

### 4.4. Western Blot Analysis

The western blot protocol was previously described in detail [25,26]. In brief, cells were harvested, washed twice in phosphate-buffered saline (PBS), and then lysed in ice-cold RIPA lysis buffer (1% Triton X-100, 1% NP40, 0.1% SDS, 0.5% DOC, 20 mM Tris-HCl pH 7.4, 150 mM NaCl, cocktail protease inhibitor, and phosphatase inhibitor cocktail) (Won-Won Biotechnology, Taipei, Taiwan) for 30 min. Lysates were boiled in 4× sample buffer dye and subjected to 10% SDS-PAGE. After separation, proteins were electrotransferred onto a nitrocellulose membrane (Amersham Pharmacia Biotech/GE, Healthcare, Piscataway, NJ, USA). The reported primary and secondary antibodies were subsequently used to probe the blots. The immunoreactive bands were detected with the enhanced chemiluminescence reaction (Millipore, Billerica, MA, USA).

### 4.5. RNA Extraction and Real-time QPCR

Total RNA was extracted from cell lines using the TOOLSmart RNA Extractor (Biotools Co., Ltd. Taipei City, Taiwan) according to the manufacturer’s instructions. RNA samples were subjected to quantitative real-time QPCR (RT-QPCR) analysis. Transcription levels were normalized to GAPDH values of each sample. Primer sequences were as follows: ERα, 5′-GGAGGGCAGGGGTGAA-3′ (sense), 5′-GGCCAGGCTGTTCTTCTTAG-3′ (antisense); NPM/B23, 5′-GGGGCTTTGAAATAACACCA-3′ (sense), 5′-GAACCTTGCTACCACCTCCA-3′ (antisense); cathepsin D, 5′-GTACATGATCCCCTGTGAGAAGGT-3′ (sense), 5′-GGGACAGCTTGTAGCCTTTGC-3′ (antisense); EBAG9, 5′-GATGCACCCACCAGTGTAAAGA-3′ (sense), 5′-AGTCAGGTTCCAGTTGTTCCAAAG-3′ (antisense); TFF1/pS2, 5′-ACATGGAAGGATTTGCTGATA -3′ (sense), 5′-TTCCGGCCATCTCTCACTAT -3′ (antisense); GAPDH: 5′-GGTATCGTGGAAGGACTCATGAC-3′ (sense), 5′-ATGCCAGTGAGCTTCCCGT-3′ (antisense). Amplifications were carried out with an initial denaturation at 95 °C for 10 min, followed by 45 cycles of 95 °C for 15 s and 60 °C for 1 min. An ABI PRISM 7900 HT instrument (Applied Biosystems, Foster City, CA, USA) was used for all reactions. Each measurement was performed in duplicate and the threshold cycle (Ct) was determined.

### 4.6. Cell Viability Assay

Approximately 1 × 10^4^ cells were seeded into each well of a 96-well culture plate for 24 h. Cells in serum-free media were treated with palbociclib, megestrol acetate, or letrozole. The MTT viability assay was then performed by adding MTT (5 mg/mL, 25 μL) into each well. After a 4 h incubation at 37 °C, the supernatant was discarded and DMSO (100 μL) was added to each well. The mixture was shaken to dissolve the formazan, and the absorbance was read at 570 nm in a multiwell spectrophotometer (VICTOR 2; Perkin Elmer GMI, Ramsey, MN, USA).

### 4.7. Clonogenic Assays

Approximately 5000 cells treated with different dose of palbociclib, megestrol acetate, or letrozole were seeded into 6-well dishes and maintained for 5−14 days. Cells were subsequently fixed in 30% methanol/12.5% acetic acid and stained with Brilliant Blue R to visualize the colonies.

### 4.8. Animals and Treatment

Six-week-old female BALB/c nude mice were obtained from the National Laboratory Animal Center, Taipei City, Taiwan. The study protocol was reviewed and approved by the Animal Care Committee of the Chang Gung Memorial Hospital Institutional Review Board (approval number 2016101703 and 2017081501). ARK2 or HEC1B cells were harvested, washed, and resuspended in Hanks’ balanced salt solution (HBSS) at 10^7^ cells/mL. Tumors were established through the subcutaneous inoculation of the cell suspension (100 μL) into the lateral hind leg of nude mice aged 6−8 weeks. After 28 days, animals were treated with palbociclib (100 mg/kg given orally 5 days a week) and/or megestrol acetate (10 mg/kg once per week). Specifically, mice were randomized to four different treatment arms (*n* = 3 each), as follows: (1) vehicle; (2) palbociclib alone; (3) megestrol acetate alone; and (4) their combination. During the course of treatment, tumor growth was monitored on a weekly basis. Tumor volume (cm^3^) was calculated as in vivo proxy of tumor mass. At the end of the experiments, tumors were excised and extracted using the RIPA buffer. Extracts were subjected to western blotting as described above.

### 4.9. Synergistic Effect of the Combined Drug Treatment

To assess the synergistic effect of palbociclib, megestrol acetate, or letrozole, we calculated the combination index (CI) using the CompuSyn software. Drug interactions were examined according to the median-effect principle of the Chou-Talalay method [35], which is based on the use of the MTT assay and the colony formation assay [36].

### 4.10. RNA Interference Procedures

Cells were transiently transfected either with specific siRNAs targeting NPM/B23, CDK4, CDK6, or control siRNA (Ambion, Austin, TX, USA) using the Lipofectamine RNAiMAX reagent (Invitrogen). In brief, Lipofectamine RNAiMAX was incubated with the Opti-MEM medium without phenol red (Invitrogen) for 5 min at room temperature. Specific siRNAs were added to the Lipofectamine RNAiMAX mixture and incubated at room temperature for 30 min to promote the formation of the transfection complex. Transfection mixtures were subsequently added to cells in the Opti-MEM medium. After 72 h, cells were harvested for subsequent PAGE and western blot analysis. The sequences of NPM/B23 siRNA were as follows: 5′-CUGGUGCAAAGGAUGAGUUTT-3′ (sense) and 5′-AACUCAUCCUUUGCACCAGTT-3′ (antisense), CAGUGGUCUUAAGGUUGAATT-3′ (sense) and 5′-UUCAACCUUAAGACCACUGTT-3′ (antisense), GGAAGAUGCAGAGUCAGAATT-3′ (sense) and 5′-UUCUGACUCUGCAUCUUCCTT-3′ (antisense), whereas the sequences of CDK4 siRNA were 5′-GAUGCGCCAGUUUCUAAGATT-3′ (sense) and 5′-UCUUAGAAACUGGCGCAUCTT-3′ (antisense), 5′-CAGAGAUGUUUCGUCGAAATT-3′ (sense) and 5′-UUUCGACGAAACAUCUCUGTT-3′ (antisense), 5′-CUCUGCAGCACUCUUAUCUTT-3′ (sense) and 5′-AGAUAAGAGUGCUGCAGAGTT-3′ (antisense), 5′-CAGCACUCUUAUCUACAUATT-3′ (sense) and 5′-UAUGUAGAUAAGAGUGCUGTT-3′ (antisense). The sequences of CDK6 siRNA were as follows: 5′-GAGUAGUUCUCUCUAACUATT-3′ (sense) and 5′-UAGUUAGAGAGAACUACUCTT-3′ (antisense), 5′-GCAGAAAUGUUUCGUAGAATT-3′ (sense) and 5′-UUCUACGAAACAUUUCUGCTT-3′ (antisense), 5′-GACUCAAGGUGGUCAGUAATT-3′ (sense) and 5′-GGAGAGUAGUUCUCUCUAATT-3′ (antisense), 5′-UUAGAGAGAACUACUCUCCTT-3′ (antisense). The sequences of negative-control siRNA were 5′-UAACGACGCGACGACGUAA-3′ (sense) and 5′-UUACGUCGUCGCGUCGUUA-3′ (antisense).

### 4.11. In Vitro Kinase Assay

Cells’ extracts were subjected to immunoprecipitation using the anti-NPM/B23 antibody. Immunocomplexes were washed with WCE buffer and kinase reaction buffer (25 mM Tris (pH 7.5), 5 mM β-glycerophosphate, 2 mM DTT, 0.1 mM Na_3_VO_4_, and 10 mM MgCl_2_). One μg of CDK6/Cyclin D1 (Invitrogen) was added to the immunocomplexes for 1 h at 30 °C. For inhibition studies, immunocomplexes and CDK6/Cyclin D1 were incubated in the presence of 2 μM palbociclib. The degree of phosphorylation was analyzed with both SDS-PAGE and immunoblotting.

### 4.12. Immunoprecipitation

Cells were harvested and washed twice in ice-cold PBS. Cell pellets were subsequently resuspended in ice-cold WCE lysis buffer (20 mM HEPES, 10% glycerol, 0.5% Triton X-100, 0.2 M sodium chloride, 1 mM EDTA, 1 mM EGTA, and protease inhibitor cocktail) for 30 min and centrifuged at 12,000 rpm at 4 °C for 30 min. Equal amounts of each cell extract protein were incubated with the indicated antibodies (2 μg) at 4 °C for 2 h. Immune complexes were captured with protein G-sepharose (30 μL; Upstate Biotechnology, Lake Placid, NY, USA) for 2 h at 4 °C under rotation. The protein G-antigen-antibody complexes were washed four times with WCE lysis buffer and boiled in 2× urea sample buffer dye (100 mM Tris-HCl, pH 6.8, 4% SDS, 0.2% bromophenol blue, 20% glycerol, 200 mM β-mercaptoethanol, 8 M urea) for subsequent PAGE and western blot analysis.

### 4.13. Immunohistochemistry and Clinical Tissue Specimens

The immunohistochemical study was approved by the local Institutional Review Board (approval number 201601370B0 and 201701722B0). Immunohistochemistry (IHC) was performed as previously described in detail [26]. In brief, 4-μm-thick formalin-fixed, paraffin-embedded tissue slices were deparaffinized in xylene and rehydrated through graded washes of ethanol in water. Sections were stained with the anti-phospho-NPM/B23 (T199) or ERα antibody using a Basic DAB detection kit (Ventana Medical Systems, Tucson, AZ, USA) in an automated IHC stainer. Hematoxylin was used for counterstaining. A semiquantitative immunostaining score (histoscore) was calculated as the percentage of positive cells multiplied by their staining intensity (0 = negative, 1 = weak, 2 = moderate, 3 = strong). Consequently, the histoscore ranged from a minimum of 0 to a maximum of 300 (i.e., 100% multiplied by 3). IHC was also performed on a commercially available endometrial cancer tissue array (EMC1021; US Biomax Inc., Rockville, MD, USA).

### 4.14. Proliferation Assay

To study the effects of NPM/B23, dephospho-mimetic NPM/B23 mutant (Thr199A) or the phospho-mimetic NPM/B23 mutant (Thr199D) mediated cell proliferation. Cells were seeded at a density of 10^4^ cells per well in complete medium for 24 h. The cells were grown for an additional 24 h after the addition of BrdU, and DNA synthesis was assayed using an enzyme-linked immunosorbent assay for BrdU incorporation (Roche Applied Science, Indianapolis, IN, USA).

## 5. Conclusions

In summary, our results demonstrated that palbociclib promoted NPM/B23 dephosphorylation at Thr199—an effect mediated by disruption of CDK6 kinase activity. This may have therapeutic relevance and can prompt further preclinical and clinical studies on the potential usefulness of palbociclib as an NPM/B23 modulator in endometrial cancer.

## Figures and Tables

**Figure 1 cancers-11-01025-f001:**
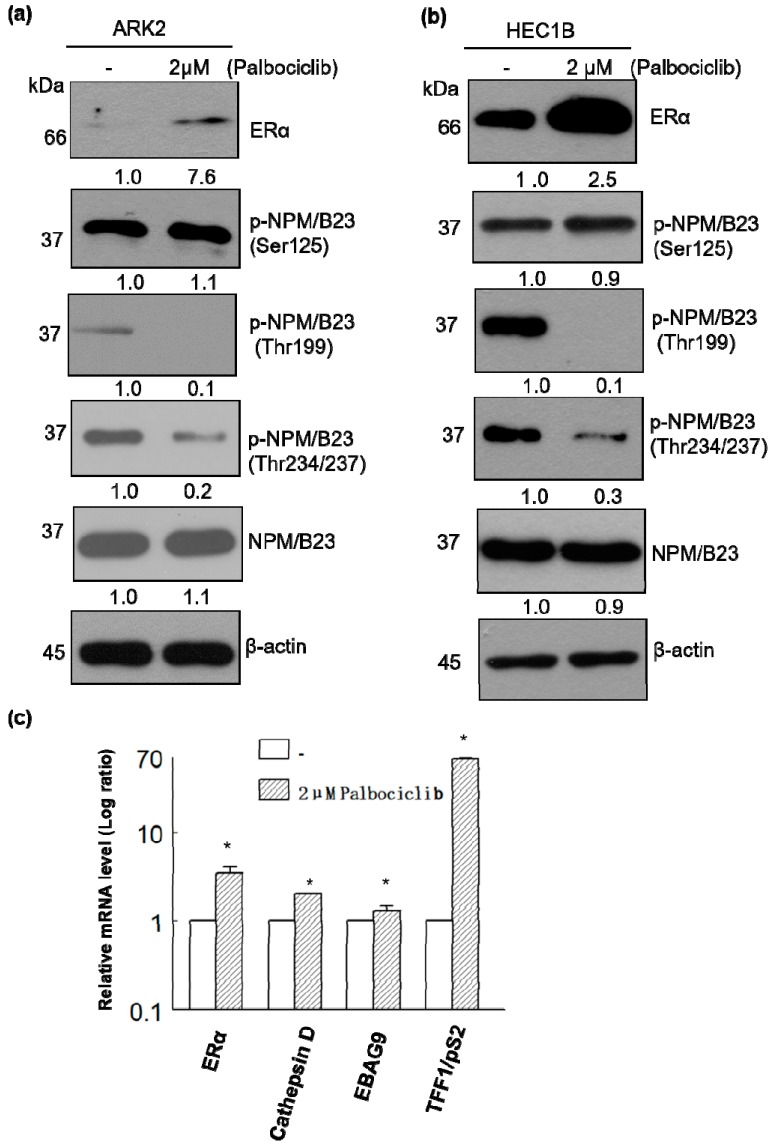
Palbociclib induces ERα expression and promotes NPM/B23 dephosphorylation at multiple sites in endometrial cancer cells. (**a**,**b**) Palbociclib (2 µM) was used to treat ARK2 (left panel) and HEC1B cells (right panel) for 24 h. Cell lysates were subsequently resolved on SDS-PAGE and subjected to immunoblotting with antibodies raised against ERα, phospho-NPM/B23 (Ser125), phospho-NPM/B23 (Thr199), phospho-NPM/B23 (Thr234/237), NPM/B23, and β-actin. Densitometry-derived values (bottom) are normalized with the control (-) that was set as 1. Data shown are derived from three independent experiments. β-actin serves as the loading control for normalization. (**c**) Palbociclib-treated ARK2 cells were collected and mRNA expression levels for ERα, cathepsin D, EBAG9, and TFF1/pS2 were analyzed using real-time qPCR (primers described in the Methods section). Data are expressed as means ± standard errors from three independent experiments. * *p* < 0.05 compared with controls.

**Figure 2 cancers-11-01025-f002:**
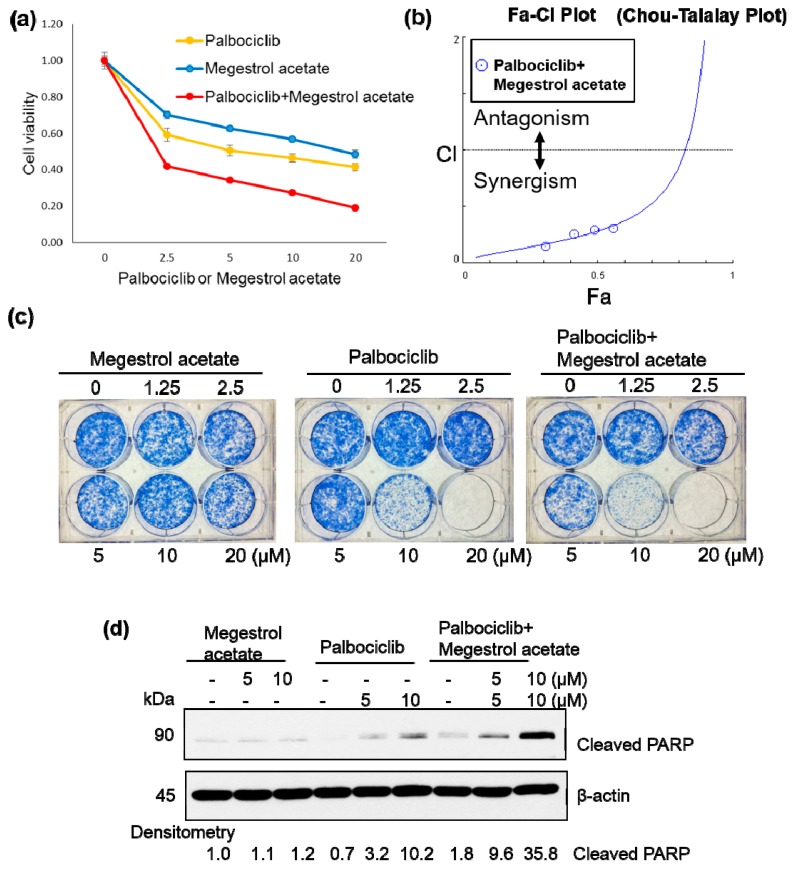
Palbociclib renders endometrial cancer cells susceptible to megestrol acetate by inducing ERα expression. (**a**,**b**) ARK2 cells were treated with vehicle (-) or different doses of palbociclib alone (0, 2.5, 5, 10, and 20 μM), megestrol acetate alone (0, 2.5, 5, 10, and 20 μM), or their combination for 72 h. Cell survival was assessed using the MTT assay. Data are expressed as fold change ± SD relative to vehicle-treated cells. All experiments were performed in triplicate (left panel). The synergistic effect of palbociclib and megestrol acetate was analyzed with the CompuSyn software (right panel). (**c**) ARK2 cells were treated with vehicle (-), different doses of palbociclib alone (0, 1.25, 2.5, 5, 10, and 20 μM), megestrol acetate alone (0, 1.25, 2.5, 5, 10, and 20 μM), or a combination (palbociclib plus megestrol acetate) for 5 days. Colony formation was analyzed with the clonogenic assay. (**d**) ARK2 cells were treated with vehicle (-), palbociclib (0, 5, and 10 μM), megestrol acetate (0, 5, and 10 μM), or a combination (palbociclib plus megestrol acetate) for 24 h. Cleaved PARP protein levels (as an index of apoptosis) were analyzed by western blotting. Values measured on densitometry (shown at the bottom) were normalized with that observed in vehicle-treated cells (set at 1). All experiments were performed in triplicate, and β-actin served as the loading control for normalization. (**e**) ARK2 were inoculated into nude mice, inhibitory effect of palbociclib plus megestrol acetate on cancer growth in a xenograft tumor model. (**f**) Extracts from tumors exposed to palbociclib, megestrol acetate, or a combination were immunoblotted with antibodies raised against ERα, phospho-NPM/B23 (Thr199), and β-actin. Densitometry-derived values (bottom) are normalized with the control that was set as 1. Data shown are derived from three independent experiments. β-actin serves as the loading control for normalization.

**Figure 3 cancers-11-01025-f003:**
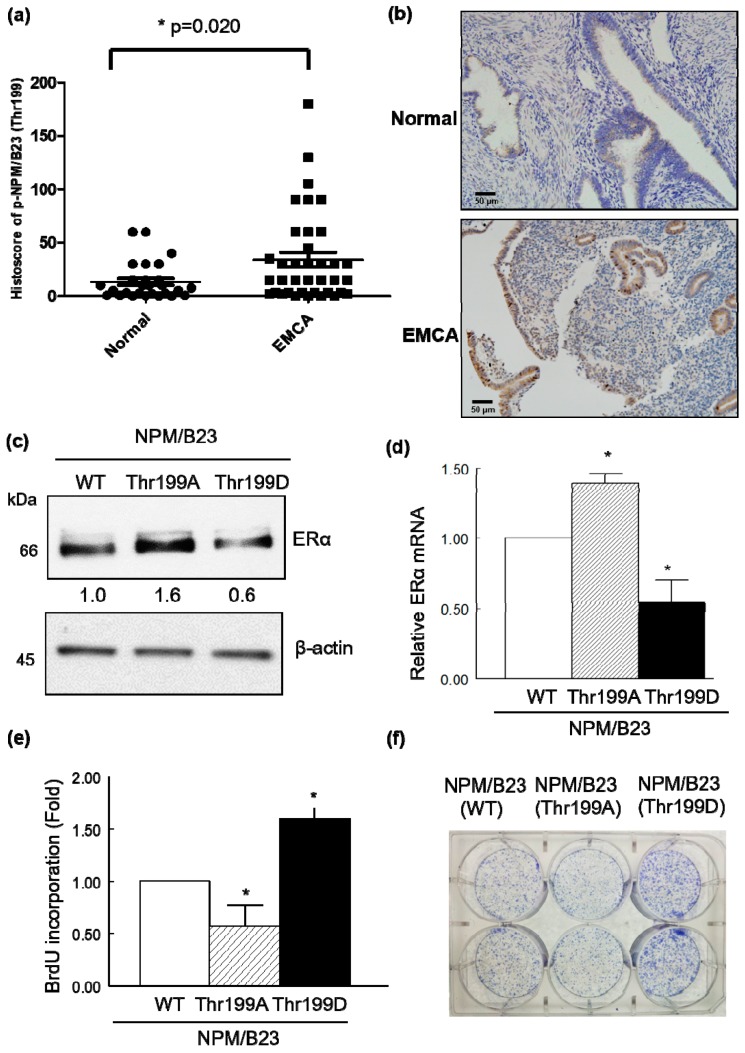
Role of phospho-NPM/B23 (Thr199) expression in endometrial tumorigenesis. (**a**) The immunohistochemical expression of phospho-NPM/B23 (Thr199) was analyzed in both normal endometrial tissues obtained from patients who underwent hysterectomy for benign gynecologic conditions (*n* = 27) and endometrial cancer (*n* = 37; Appendix A). (**b**) Representative immunohistochemical staining of phospho-NPM/B23 (Thr199) in benign endometrial tissue (upper panel) and endometrial cancer (EMCA; lower panel). Magnification: 100×; scale bar: 50 μm. (**c**) ERα-positive HEC1B cells were transfected with NPM/B23 (WT), the dephospho-mimetic NPM/B23 mutant (Thr199A), or the phospho-mimetic NPM/B23 mutant (Thr199D) for 24 h. Cell lysates were resolved on SDS-PAGE and subjected to immunoblotting with antibodies raised against ERα and β-actin. Densitometry-derived values (bottom) are normalized with the NPM/B23 (WT), which was set as 1. Data shown are derived from three independent experiments. β-actin serves as the loading control for normalization. (**d**) HEC1B cells were transfected with NPM/B23 (WT), the dephospho-mimetic NPM/B23 mutant (Thr199A), or the phospho-mimetic NPM/B23 mutant (Thr199D) for 24 h. RT-qPCR was used to analyze mRNA expression levels using the reported primers. * *p* < 0.05 versus control. Experiments were performed in triplicate and data are expressed as means ± standard errors of the mean. (**e**,**f**) ARK2 cells were transfected with the NPM/B23 (WT), dephospho-mimetic NPM/B23 mutant (Thr199A), or the phospho-mimetic NPM/B23 mutant (Thr199D) for 24 h. For cell proliferation assay, cells were seeded at a density of 10^4^ cells per 96 well in complete medium for 24 h. The cells were grown for an additional 24 h after the addition of BrdU. For colony formation, the cells were seeded at a density of 5000 cells per 6 wells in complete medium for 6 days. Data are expressed as means ± standard errors from three independent experiments. * *p* < 0.05 versus control.

**Figure 4 cancers-11-01025-f004:**
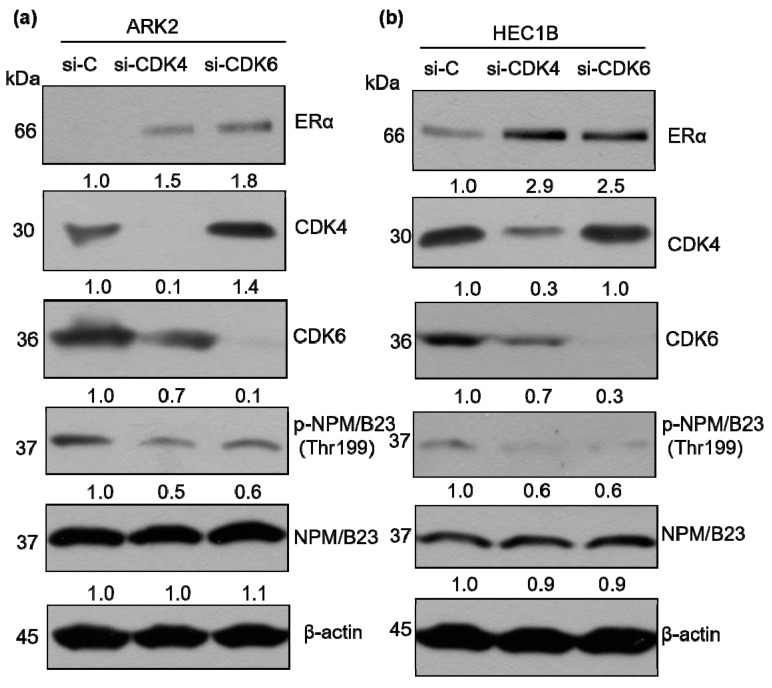
CDK6-mediated phosphorylation of NPM/B23 promotes ERα expression. (**a**,**b**) ARK2 or HEC1B cells were transiently transfected with control siRNA (si-C), CDK4 siRNA (si-CDK4), or CDK6 siRNA (si-CDK6) for 72 h. Cell lysates were subjected to western blot using antibodies raised against ERα, CDK4, CDK6, phospho-NPM/B23 (Thr199), NPM/B23, and β-actin. Densitometry-derived values (bottom) are normalized with the control (si-C), which was set as 1. Data shown are derived from three independent experiments. β-actin serves as the loading control for normalization.

**Figure 5 cancers-11-01025-f005:**
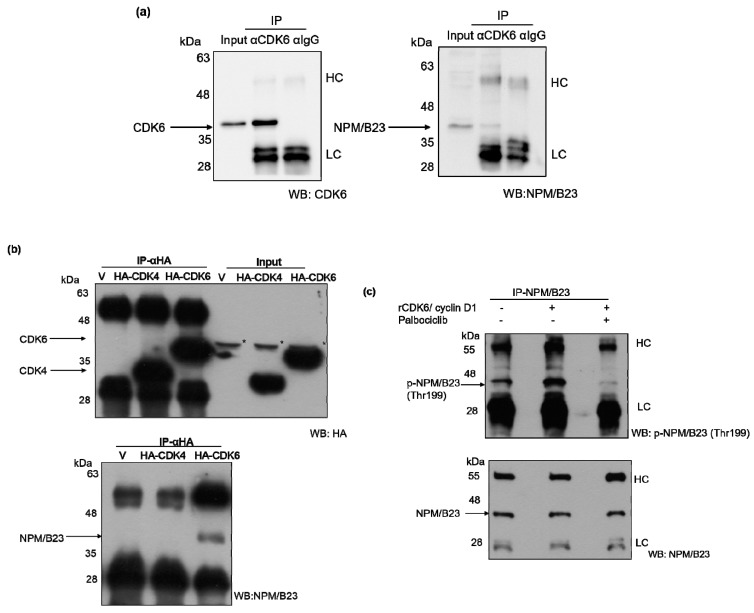
CDK6-mediated NPM/B23 hyperphosphorylation. (**a**) ARK2 whole-cell lysates were immunoprecipitated (IP) with an anti-CDK6 antibody (α-CDK6) and subsequently analyzed by immunoblotting with an anti-CDK6 antibody or an anti-NPM/B23 antibody. A control IgG antibody (α-IgG) was used for mock immunoprecipitation. (**b**) ARK2 cells were transiently transfected with pcDNA (V), HA-CDK4, or HA-CDK6-expressing vectors for 24 h. ARK2 whole-cell lysates were immunoprecipitated (IP) with an anti-HA antibody (α-HA) and subsequently analyzed by immunoblotting using an anti-HA antibody or an anti-NPM/B23 antibody. The asterisks indicate non-specific bands. (**c**) Lysates from ARK2 cells underwent immunoprecipitation using an anti-NPM/B23 antibody. An in vitro kinase assay was performed on immunoprecipitates using a recombinant rCDK6/cyclinD1 kinase (as described in the Methods section). The resulting products were probed with the indicated antibodies (HC, heavy chain; LC, light chain).

**Figure 6 cancers-11-01025-f006:**
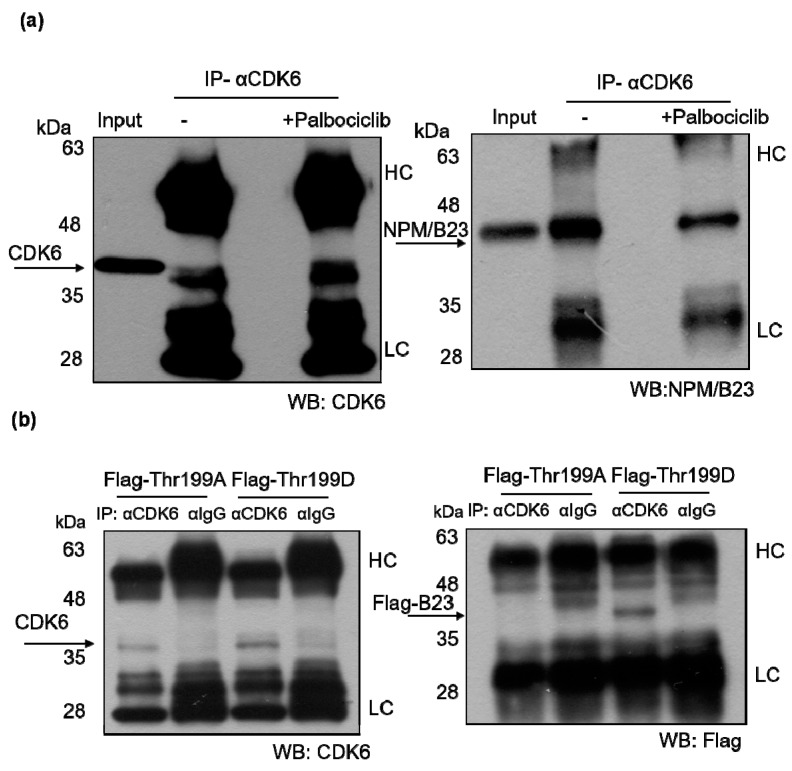
Phosphorylated NPM/B23 (Thr199) is capable of associating with CDK6. (**a**) ARK2 were treated with 2 μM palbociclib for 24 h (at the experimental conditions described in the Methods section). Whole-cell lysates were immunoprecipitated (IP) with an anti-CDK6 antibody (α-CDK6) and subsequently analyzed by immunoblotting with an anti-CDK6 antibody or an anti-NPM/B23 antibody. (**b**) ARK2 cells were transfected with the dephospho-mimetic NPM/B23 mutant (Thr199A) or the phospho-mimetic NPM/B23 mutant (Thr199D) for 24 h and immunoprecipitated (IP) with an anti-CDK6 antibody (α-CDK6). Subsequently, immunoblotting with an anti-CDK6 antibody or an anti-NPM/B23 antibody was performed. A control IgG antibody (α-IgG) was used for mock immunoprecipitation (HC, heavy chain; LC, light chain).

**Figure 7 cancers-11-01025-f007:**
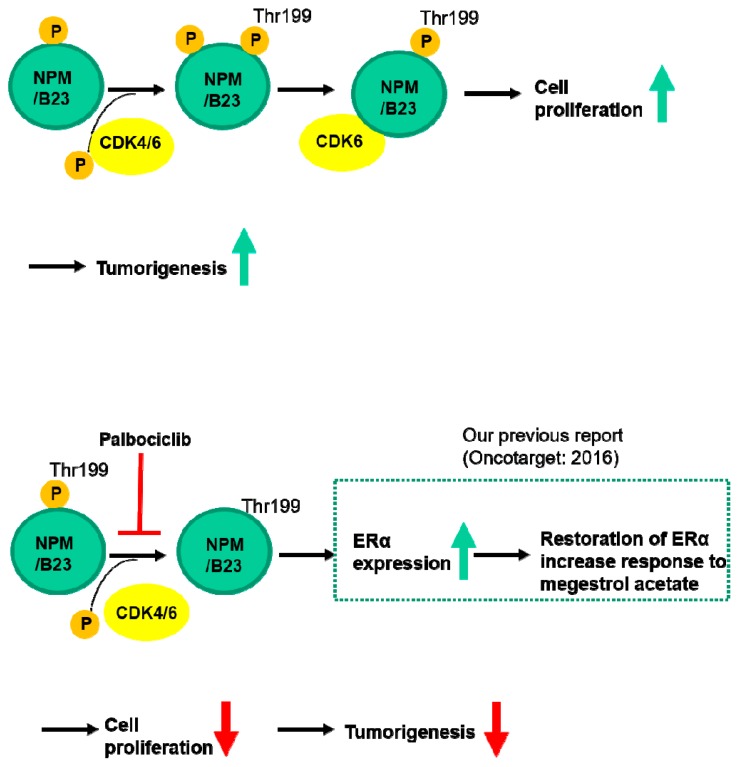
Schematic representation of the inhibitory effects of palbociclib on endometrial cancer cell growth. Palbociclib is capable of inhibiting phosphorylation of NPM/B23 (Thr199) and promoting ERα expression, ultimately sensitizing hormone-refractory endometrial cancers to endocrine therapy.

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
