# Peer review of "Palbociclib Promotes Dephosphorylation of NPM/B23 at Threonine 199 and Inhibits Endometrial Cancer Cell Growth"

_cancers, 2019, doi:10.3390/cancers11071025_

Round 1

Reviewer 1 Report

This manuscript revealed the effect of CDK4/6 inhibitor via NPM/B23 and the effect of megestrol via ERα up-regulation in endometrial cancer cell lines. The results are interesting and most points are revised properly.

Reviewer 2 Report

The authors have thoroughly edited the manuscript and is greatly improved.

This manuscript is a resubmission of an earlier submission. The following is a list of the peer review reports and author responses from that submission.

Round 1

Reviewer 1 Report

This paper is described about the potential of new therapy against endometrial cancer. These data are very interested. It may be good therapy for ER positive endometrial cancer.

Palbociclib and megestrol acetate synergistically inhibit survival. The data shown ER positive in ARK2 and HEC1B, but not shown PgR. How effects these combination therapy against ER negative endometrial cells, ex uterine serous carcinoma. 

Reviewer 2 Report

1) Page 2, lines 78-80: Is this description correct? Can NPM/B23 silencing activate ERα expression?

2) In figure 2, ERα-negative cell line was not affected by megestrol and palbociclib. Why does the combination of megestrol and palbociclib decrease tumor growth? Is this effect only via ERα, or is there any other mechanism? 

I doubt that activation of ERα by palbociclib might lead to growth inhibition by megestrol.

Does the synergistic effect disappear if the effect via ERα was blocked by estrogen antagonist or by silencing? Please discuss this aspect in the discussion section.

3) In line with question 2, have the authors checked progesterone receptor (PgR) expression? Effect of megestrol mainly manifests via PgR, rather than ERα.

4) In supplementary figure 2, combination of letrozole with palbociclib had synergistic effects against tumor growth. Why does letrozole have synergistic effect with palbociclib? Is this effect independent of ERα expression? Is there a different pathway related to megestrol administration?

5) The dose of drugs was described in figure 2 legend; however, the data corresponding to the dose of 1.25 degree have not been described in figure 2a. 

6) Page 7, line 187: The author described “transfection with the phosphor-mimetic NPB/B23”. Is “transfection with the dephospho-mimetic NPB/B23” correct?

Reviewer 3 Report

Palbociclib is a cyclin-dependent kinase (CDK) 4/CDK6 inhibitor approved for breast cancer that is estrogen receptor (ER)-positive and human epidermal growth factor receptor 2 (HER2)-negative. Authors showed that Palbociclib-induced NPM/B23 dephosphorylation and sensitized endometrial cancer cells to megesterol acetate through upregulation of ERα expression. Here are several questions that need the authors to clear up

1. Authors showed Palbociclib increased ER expression in HEC-1B cells and activated ER expression in ARK2 cells. This is surprised results. Did authors exam other ER negative cell lines? Do you have any literatures to support your results?

2. Did authors detect PR expression in both cell lines? Megesterol acetate mainly depends on PR expression to inhibit cell proliferation in cancer cells.

3. In Fig2d, authors should include PRAP expression in western blotting

4. In Fig3E, authors should use tumor volume instead of fold change

Reviewer 4 Report

This manuscript is interesting and overall well written. The results presented follow an overall logical succession of experiments. This study provides important findings linking CDK4/6 inhibition, NPM phosphorylation status on T199 and ERa expression, which may provide a mechanism to re-sensitise ERa negative tumours for hormone-mediated therapy.

There are several areas for improvement and the following issues should therefore be addressed.

1. This study addresses both oncologists and molecular biologists/cell biologists who may have different knowledge on therapies in endometrial cancer. Background information on ERa should be provided and in the introduction including information on the clinical consequence of the lack or ERa. The importance of the upregulation of its levels should also be explained. This information should be linked to the rational for testing CDK inhibitors and NPM phosphorylation. 

2.  Quantification of Western immunoblotting should be provided, in figure 1a-b, figure 3c. figure 4a-b.

3. Figure 2c:  the conclusion of synergism by both compounds is not obvious. Quantification should be provided for these results and for all in other figures.

4. The number of experiments (biological replicates) should also be stated in figure legends.

5. Figure 3e-f. The results from a WT control needs to be shown. It should not be possible to draw any conclusions from the effect of the T199D and T199A mutants without it.

6. Figure 5. Right panels of a and b. The resulting band identifying NPM in the CDK6 IPs is quite different, a thin band in panel a corresponding to the lower form in the input (un-modified?) compared to thick band in panel b. Please explain.

Additional minor comments:

·       Line 79: what does AP2g stands for?

·       Line 85: please provide a brief description of megesterol acetate. What is this compunt usually used for?

·       Line 126-8: please provide a brief description of letrozol. State what is this usually used for and what is the rational for using it in this study.

·       Line 95: please provide the rational for the choice of these 2 cell lines in relation to their ERa expression status

·       Line 100: human gene names should be written in italic

Figures

·       Figure 1a-b: provide the symbol for micro

·       Figure 2f: move this panel to the right

·       Figure 2e legend: specify which cell line was used for the xenograft

·       Figure 3c. add the standards mW

·       For clarity, I would recommend to reorganise figures 4 and 5, as well as supp fig 5 and show Fig 5a, supp Fig5 together with Fig 4c  to follow the flow of the main text, such as:

Fig 5a remains Fig 5a, supp Fig 5 becomes Fig 5b and Fig 4c becomes Fig 5c.

Fig 5b and c would then become Fig 6a and Fig6b respectively.

·       For all the figures showing colony formation data, specify which well corresponds to which concentration of each of the compounds used.